# Intranasal and Serum Gentamicin Concentration: Comparison of Three Topical Administration Protocols in Dogs

**DOI:** 10.3390/vetsci10080490

**Published:** 2023-07-28

**Authors:** Tom Biénès, Aurélie Lyssens, Hélène Machiels, Marie Eve Hercot, Aline Fastres, Tutunaru Alexandru-Cosmin, Marine Deville, Corinne Charlier, Frédéric Billen, Cécile Clercx

**Affiliations:** 1Department of Clinical Sciences, Faculty of Veterinary Medicine, B67 Sart Tilman, University of Liege, 4000 Liege, Belgium; aurelie.lyssens@uliege.be (A.L.); helene.machiels@uliege.be (H.M.); hercot_me@hotmail.com (M.E.H.); actutunaru@uliege.be (T.A.-C.); fbillen@uliege.be (F.B.); 2Laboratory of Clinical, Forensic, Industrial and Environmental Toxicology, Center for Interdisciplinary Research on Medicines (CIRM), University Hospital of Liege, 4130 Liege, Belgium; m.deville@chuliege.be (M.D.); c.charlier@chu.ulg.ac.be (C.C.)

**Keywords:** topical administration protocol, gentamicin, dogs, antibiotic, nasal disease

## Abstract

**Simple Summary:**

The topical administration of antibiotics for the treatment of nasal cavity and frontal sinus infections has not been assessed in dogs. In ten healthy beagle dogs, we administered gentamycin by inhalation twice daily for 3 min (3-min protocol) and 10 min (10-min protocol), as well as by drop administration (drop protocol) twice daily, for one week each. We found that the gentamicin concentration in the nasal lavage fluid (NALF) was always effective and was the highest after the drop protocol, while it was always below the toxic dosage in the serum. The drop protocol appears the most adequate method to administer gentamicin for nasal topical treatment.

**Abstract:**

Antimicrobials’ topical administration efficacy has not been assessed in dogs with upper respiratory tract disease. The aim was to compare the concentration of gentamicin in nasal lavage fluid (NALF) and in serum after three topical protocols. This was a prospective crossover study of ten healthy dogs. Gentamicin was nebulized for a duration of 1 week, twice a day, for 10 min in the first protocol (10-min protocol) and for 3 min in the second protocol (3-min protocol), while the third protocol consisted of the administration of 0.25 mL of gentamicin in each nostril (drop protocol). Median concentrations of gentamicin in NALF were 9.39 µg/mL (8.12–19.97 interquartile range), 4.96 µg/mL (4.60–6.43) and 137.00 µg/mL (110.5–162.00) in the 10-min protocol, 3-min protocol and drop protocol, respectively. The result for the drop protocol was significantly higher than those of both nebulization protocols in NALF (*p* = 0.039). In serum, the gentamicin concentration was 0.98 µg/mL (0.65–1.53) and 0.25 µg/mL (0.25–0.44) in the 10-min and 3-min protocols, respectively. Gentamicin was not detected in the serum of seven out of ten dogs in the drop protocol, and gentamicin was significantly higher in the 10-min protocol compared to the drop protocol (*p* = 0.001). This study found that the 10-min, 3-min and drop protocols achieved superior concentrations in NALF compared to the minimum inhibitory concentration for gentamicin-sensitive bacteria, while remaining below the toxic values in blood.

## 1. Introduction

Nasal disease is frequently encountered in dogs. Nasal neoplasia, lymphoplasmacytic rhinitis, fungal rhinitis, nasal foreign bodies and periodontal disease are the most common etiologies of rhinosinusitis in dogs [1,2,3,4]. Primary bacterial rhinosinusitis is controversial in veterinary medicine; however, bacterial infection can frequently occur as a result of any underlying nasal condition [1,2,3]. In human medicine, independently of the primary etiology, an inflammatory insult decreases mucociliary clearance and promotes bacterial colonization, playing a role in the pathogenesis of nasosinusal diseases [5]. Adequate treatment is crucial to avoid secondary induced irreversible lesions of the turbinates and mucosal remodeling [6,7]. In general practice, dogs with nasal disease are often treated with systemic courses of oral non targeted antibiotics as first intention. However, repeated empirical administration of systemic oral antibiotics is related to development and spread of drug resistant microorganisms [8]. The increased prevalence of antibiotic resistance has become a major concern in recent decades [8].

In humans with upper respiratory tract diseases, the topical administration of antimicrobials has become increasingly popular [8]. Inhalations and nasal drops are two methods of treatment administration providing direct delivery to the sinonasal cavity tissues while minimizing the risks of toxicity related to systemic therapy [8,9,10]. The high concentration of antibiotics at the site of infection also enables the effective eradication of the bacterial biofilm [11].

Topical hydrophilic antibiotics such as aminoglycosides are of particular interest in order to reduce bacterial resistance [8]. Topical gentamicin targets organisms that might be resistant to oral antibiotics, while resistance to prolonged topical administration is mild [8,9]. Antimicrobials such as b-lactams achieve maximum bacterial killing when the drug concentration remains constantly above the minimum inhibitory concentration (MIC) [12]. For aminoglycosides drugs, the rate and extent of bactericidal activity depend on the concentration of the drug, which is referred to as concentration-dependent killing [9,13]. The efficacy of gentamicin has been linked to the ratio of the maximal concentration (C_MAX_) divided by the MIC (C_MAX_/MIC) of the pathogen, the optimum bactericidal activity being achieved when the peak concentration to MIC ratio is high (>10 µg/mL) [13,14,15]. The MIC for gentamicin-sensitive bacteria is between 2 and 4 µg/mL [16]. The toxic gentamicin concentration in serum has been reported as 2–4 µg/mL [17]. In human medicine, the pharmacodynamics of topical gentamicin in the sinonasal cavity and serum have not been widely investigated and there is no consensus regarding the optimal use of this administration route [8,9,10,17,18]. In dogs, to the best of our knowledge, there are no such studies available. In dogs, the most commonly described intranasal delivery methods relate to nebulization of steroids or antibiotics for management of lower airway disease [19,20]. Since nasal cavity is a suitable target for systemic delivery of drugs due to a large mucosal surface area combined with extensive vascularization providing an optimal absorption surface for drug delivery, intranasal delivery of drugs, such as sedatives or vaccines is also used to induce robust and prompt systemic effect [20,21,22,23]. However local administration of antibiotics is not commonly used in dogs for treatment of upper respiratory tract diseases.

Nasal lavage (NAL) collection has been used in humans to investigate and monitor rhinitis [24,25,26]. This method is considered reliable and less invasive compared with other current diagnosis techniques such as histopathology [27]. Furthermore, this technique has been described as a collection method in epidemiologic and experimental studies [24]. In authors’ knowledge, the reproducibility and cytologic description of nasal lavage fluid (NALF) collection protocols have not been described in dogs.

The aim of this study was to compare the concentrations of gentamicin in NALF and in serum obtained after three different protocols of topical administration, including a 10-min nebulization delivery protocol, a shorter (3-min) one, and an intranasal drop deposit protocol, in healthy beagle dogs. For this purpose, we also considered a standardized procedure of nasal lavage (NAL).

## 2. Materials and Methods

### 2.1. Dog Population 

Ten healthy experimental beagle dogs (10 males), aged 2 to 15 years, with a median of 6 years [4,5,6,7,8,9,10,11,12] (median (25th percentile–75th percentile)), with a body weight between 15.1 and 18.7 kg (17.1 (15.6–18.6)), were included in the experimental study, which was approved by the Ethical Committee of the University of Liege (protocol #2299). At inclusion, the dogs were considered healthy based on the absence of clinical signs, normal physical examination and normal blood hematology and serum biochemistry. 

### 2.2. Protocol

This prospective crossover study was conducted over a 42-day period. In each dog, a NAL procedure was performed under anesthesia at four different timepoints. The first NAL procedure was performed before gentamicin administration at day 0. Then, three further NAL procedures were performed following weekly gentamicin administration using three different topical gentamicin administration protocols, at days 14, 28 and 42. Each protocol was separated by a one week wash-out period (Figure 1).

For the two gentamicin nebulization administration protocols, gentamicin (GENTA-kel 5%^®^; KELA, Hoogstraten, Belgium) was nebulized using an ultrasonic nebulizer (Aeoroflaem^®^, Martino, Desenzano del Garda, Italy). Ultrasonic nebulization induces the water vibration of the gentamicin solution, allowing the solution to mix with the air. Nebulization was performed for 10 min, twice a day, for 1 week in the first protocol (10-min protocol) and for 3 min, twice a day, for 1 week in the second protocol (3-min protocol). The 10-min protocol and 3-min protocol resulted in the nebulization of a total amount of 1.9 mL (95 mg) and 0.57 mL (28 mg) of gentamicin, respectively. The third protocol consisted of the administration of 0.25 mL of gentamicin (12.5 mg) in each nostril, twice a day, for 1 week (drop protocol). Gentamicin was administered at the same concentration (50 ± 0.01 mg/mL) for the three different protocols, to reduce the effect of the initial dose on the drug concentration. NALF was collected as quickly as possible after the end of the last inhalation in both the 10-min and 3-min protocols and the last intranasal administration in the drop protocol, in order to reflect the maximal concentration (C_MAX_) obtained after topical administration. 

### 2.3. Sample Collection 

Dogs were anaesthetized using butorphanol (0.2 mg/kg; Butomidor^®^, Richter Pharma, silures, Autria) in combination with medetomidine (5 μg/kg; Medetor^®^, CP-Pharma, Burgdorf, Germany) intravenously. Propofol (2–4 mg/kg to effect; Propovet^®^, Zoetis, Malakoff, France) intravenously was used for induction. Anesthesia was maintained with isoflurane (Iso-Vet^®^; Eurovet, Bladel, Netherlands). Dogs with intra-tracheal tubes were placed in ventral recumbency; the nasopharynx was manually obstructed. A 4-cm-long 12-Fr fenestrated catheter connected to a 60-mL syringe was introduced in the first third of the left nasal cavity. Twenty mL (20 ± 1 mL/kg) of sterile isotonic saline solution was then injected, with the ipsilateral nostril being completely manually obstructed to prevent leakage of the solution. The fluid was then directly aspirated in the syringe using manual suction while progressively removing the catheter from the nasal cavity (Appendix A). The time of the procedure was recorded. Venous blood samples were obtained after the NALF procedure by jugular venipuncture using 5-mL plastic syringes and 21-G needles and placed immediately into 5-mL dry tubes.

### 2.4. Sample Processing

The volume and macroscopic aspects (clear, yellowish, hemorrhagic or lactescent) of NALF samples were recorded. For each NALF sample, at days 0, 14, 28 and 42, one aliquot (1 mL) was used for both total cell count calculation using a hemocytometer on 100 cells and for cytospin preparations (centrifugation at 221 g, for 4 min at 20 °C, Thermo Shandon Cytospin©4). Cytospin preparations were stained with May–Grünwald–Giemsa stain and were used to assess differential cell counts. The total cell count and differential cell count were established by two independent experienced veterinarians. 

At days 14, 28 and 42, each NALF sample was centrifuged at 35,000 rotations per minute for 5 min. Then, a 1-mL aliquot of NALF supernatant was transferred into cryotubes and stored at 6 °C. Venous blood was centrifuged at 35,000 rotations per minute for 5 min. Serum was then placed in cryotubes and stored at 6 °C until batched analysis. 

### 2.5. Gentamicin Concentration 

The gentamicin concentration was measured for all dogs in NALF and serum on the same day as the collection, at each timepoint, at the Laboratory of Toxicology (Center for Interdisciplinary Research on Medicines, University Hospital of Liege, Liege, Belgium). Gentamicin quantification was performed via a Particle-Enhanced Turbidimetric Inhibition Immunoassay on an Alinity analyzer [28]. Using this technique, the lower limit of quantification (LOQ) of gentamicin was validated on human serum samples at 0.5 µg/mL [19]. Values below LOQ were assimilated as LOQ/2 (0.25 µg/mL) for statistical analysis. In the absence of detection, the values were zero.

### 2.6. Statistical Analysis

All statistical analyses were performed with the XLSTAT software (version 2022.4.1) for Windows. Normality was checked with Shapiro–Wilk tests in all groups. Data on the serum and NALF gentamicin concentrations were not normally distributed. Data results were expressed as the median and interquartile range.

The gentamicin concentrations in each different protocol were compared in the serum and NALF using the Friedman test and Nemenyi’s procedure for pairwise comparisons. Concentrations in serum and NALF were compared for each protocol using the Wilcoxon signed rank test. For all tests, a *p*-value lower than 0.05 was considered significant.

## 3. Results

### 3.1. Nasal Lavage and Serum Sampling 

The median time of the whole NAL procedure was 32 s (range 24–41 s). NAL was performed within 15 to 23 min (median = 17) of the last gentamicin administration. The volume of NALF recovered varied between 5 and 16 mL, with a median value of 11 mL, representing 56% of the instilled liquid. The NALF appeared clear to slightly lactescent in all dogs. The NAL sampling technique was well tolerated and no adverse side effect was observed during or after the procedure. 

Serum was sampled within 27 to 36 min (median of 33) of the last gentamicin administration.

### 3.2. Total Cell Counts and Differential Cell Counts in Nasal Lavage Fluid

At day 0 before gentamicin administration in the ten dogs, the total cell count of NALF ranged from 0 to 60 cells/µL (10 cells/µL (0–20), median (interquartile range 25–75)). Three types of cells were observed in the differential cell count: squamous epithelial cells (50% (38–52)), epithelial cells (48% (48–55)) and neutrophils (2% (0–4)). At day 14 after the 10-min protocol, the total cell count of NALF was 8 cells/µL (0–18). The differential cell count included epithelial cells at 58% (48–63), squamous epithelial cells at 40% (32–52) and neutrophils at 2% (0–4). At day 28 after the 3-min protocol, the total cell count of NALF was 10 cells/µL (0–18). The differential cell count included epithelial cells at 62% (48–65), squamous epithelial cells at 48% (33–54) and no neutrophils. At day 42 after the drop protocol, the total cell count of NALF was 15 cells/µL (0–19). The differential cell count included epithelial cells at 50% (40–65), squamous epithelial cells at 49% (35–55) and neutrophils at 1% (0–1). No significant difference was present between the total cell count and differential cell count in the three protocols.

Squamous epithelial cells were characterized by large cells of polyhedral and angular appearance. The nucleus was either still present, pycnotic or absent and represented less than 15% of the cell diameter. Squamous epithelial cells had numerous keratohyalin granules in the cytoplasm (Figure 2). Epithelial cells were present singly or in small clusters, identified with a rounded shape. They had a large, round nucleus and little cytoplasm, with a nuclear diameter representing more than 30% of the cell diameter. Cilia, if present, were located opposite to the nucleus (Figure 2). No bacterial contamination was observed.

### 3.3. Gentamicin Concentration in Nasal Lavage Fluid and Serum

The median concentration of gentamicin in NALF was not significantly higher in the 10-min protocol (9.39 µg/mL (8.12–19.97)) compared with the 3-min protocol (4.96 µg/mL (4.60–6.43)) (*p* = 0.175). However, the median concentration of gentamicin in NALF in the 10-min protocol was approximately two (1.9) times higher than in the 3-min protocol. The NALF gentamicin concentration was significantly higher in the drop protocol (137.00 µg/mL (110.5–162.00)) compared with both the 10-min protocol and the 3-min protocol (9.6 times higher and 20.7 times higher, respectively) (Figure 3).

The median serum gentamicin concentration in the 10-min protocol was 0.98 µg/mL (0.65–1.53). Serum gentamicin was detected in all dogs with the 10-min protocol, but was below the LOQ in one dog. In the 3-min protocol, the median concentration of gentamicin was 0.25 µg/mL (0.25–0.44); the gentamicin concentration was below the LOQ in five dogs and undetectable in two dogs. With the drop protocol, gentamicin was not detected in the serum of seven out of ten dogs, while the serum gentamicin concentration was below the LOQ in two dogs. The serum concentration of gentamicin was significantly higher in the 10-min protocol compared to the drop protocol but not the 3-min protocol (*p* = 0.001). (Figure 4).

Finally, the calculated gentamicin concentration NALF/serum ratio was below 20 in the inhalation protocols (at 13 in the 10-min protocol and 19 in the 3-min protocol), while it was above 1000 in the drop protocol.

## 4. Discussion

This study assessed the concentration of gentamicin in NALF and serum after three different one-week topical administration protocols in healthy dogs. NALF gentamicin concentrations were always superior to the reported minimal effective dose for aminoglycoside-sensitive bacteria and was the highest after the drop protocol. In the serum, gentamicin was below the toxic dosage in all dogs. The drop protocol offered the highest gentamicin NALF concentration combined with the lowest serum concentration and appears therefore to be the most suitable method for the administration of gentamicin for nasal topical treatment.

The three topical protocols investigated in the present study appear to be good alternatives to systemic administration, enabling a high concentration in NALF combined with a low concentration in serum. All three topical administration protocols resulted in gentamicin concentrations in NALF above the MIC, even without taking into account any dilution effect associated with the NAL procedure. Indeed, an MIC dose for gentamicin-sensitive bacteria has been reported between 2 and 4 µg/mL [13,16]. As mentioned in the Introduction, the rate of bacterial killing of aminoglycosides depends on the peak concentration of the drug, while other antimicrobials such as b-lactams achieve maximum bacterial killing when the drug concentration remains constantly above the MIC [13]. This type of killing is therefore referred to as concentration- or dose-dependent killing. A high peak gentamicin concentration relative to the MIC for the infecting organism is a major determinant of the clinical response to therapy [13]. Moreover, since the values were not corrected for any dilution effect, the real concentrations in the nasal epithelial lining fluid (ELF) can be expected to be much higher than the ones measured in NALF. Therefore, we expect that the concentrations obtained after the three protocols are effective against gentamicin-sensitive bacteria growing in the nasal cavities.

In the serum, gentamicin was not always detectable, especially in the drop protocol. When detected, its concentration was often below the quantification limit of the test. In all dogs in this study, it was always lower than the reported toxic blood concentration. Furthermore, the dose administered topically in all three protocols was lower than the dose accepted intramuscularly (6 mg/kg q 24 h during 5 days), equivalent to 102 mg gentamicin for a 17 kg beagle [29]. Indeed, it is accepted that peak concentrations above 2 µg/mL in serum are a risk factor for nephrotoxicity and ototoxicity, while the peak toxic concentration is above 12 µg/mL [16,17]. Since gentamicin exhibits a narrow range between toxic and therapeutic dosages, a guarantee of a low concentration in the serum is a requirement before recommending any therapeutic protocol using this drug. In men, gentamicin concentration values reported in serum after gentamicin nasal irrigation were in a range similar to the one of the present study (0.42 µg/mL range 0.3 to 0.7 µg/mL in patients with detectable levels) [10]. In men, the peak serum concentration after nasal irrigation was described after 30 to 60 min [30]. Therefore, although we measured the concentration of gentamicin in serum only once, 35 min after drug administration, we can assume that the gentamicin levels measured in the serum of dogs in the present study were close to the peak serum concentration and did not exceed toxic values.

In veterinary medicine, a prolonged 3- to 6-week protocol of 10-min inhalations of 5% gentamicin twice daily, similar to the 10-min protocol used in the present study, has been described to successfully treat dogs with pulmonary bordetellosis, with no reported toxicity to the kidney [20]. The risk of toxicity of the three protocols investigated here in normal dogs appears therefore negligible. In humans, approximately 3% of orally administered aminoglycosides are absorbed [10]. However, the mechanism by which gentamicin enters the serum after topical therapy remains unclear. Potential pathways include translocation across mucous membranes or gastrointestinal absorption of the swallowed drug. Given the relatively high volume of administration, it is possible that the gastrointestinal route contributed to the serum levels measured.

Despite the lower dosage administered in the drop protocol compared to both inhalation protocols, it allowed a concentration in NALF that was 9.6 and 20.9 times higher than that of the 10-min protocol and the 3-min protocol, respectively. In addition, the same protocol resulted in a lower mean serum concentration. The calculation of a higher NALF gentamicin /serum ratio in the drop protocol compared with the inhalation protocols shows that this protocol may be the most appropriate. When looking at the feasibility of the protocols, the drop protocol was faster and better tolerated compared to the two inhalation protocols. Furthermore, the drop protocol does not require special equipment, such as a nebulizer. For all these reasons, drop administration appears to be the best topical technique while limiting the systemic absorption of gentamicin.

We believe that these three protocols can offer alternatives to systemic antibiotic treatment. However, since the present study was conducted on healthy animals (free of nasal disease), the efficacy of local protocols still needs to be investigated in dogs with acute or chronic rhinosinusitis. In humans, gentamicin nasal irrigation has been described as a successful treatment for chronic rhinosinusitis, especially in cases of multi-drug-resistant bacteria [18,31]. Although a study described the success of gentamicin topical treatment in dogs with lower airway infectious diseases, there is a lack of research documenting the use of such protocols in dogs with nasal diseases [20].

In human medicine, nasal lavage collection and analysis is being used as a minimally invasive tool to investigate and monitor rhinitis [30,31]. This technique has also been described as a collection method in epidemiologic and experimental studies, such as the determination of inflammatory cytokines in relation to allergic rhinitis or drug measurement [24,25,30]. The procedure used in the present study was quick, cheap, minimally invasive and reproducible, and it yielded adequate NALF samples in all dogs. It showed that in healthy dogs, NALF has poor total cellularity and is mainly composed of squamous and epithelial cell populations. These results are consistent with the composition of the nasal epithelium, which is a multi-layered nasal and keratinized pigmented epithelium, and are also consistent with findings in human medicine [32,33]. However, the diagnostic and prognostic value of NALF analysis in dogs with nasal disorders has to be determined. The detection of inflammatory cells, fungal and/or bacterial infectious agents or tumor cells could help in the diagnosis of nasal pathologies. One of the major expected complications would be the contamination of NALF with blood. In this study, the total cell count and differential cell count did not differ at different timepoints. Therefore, the lack of significant change suggests the absence of a nasal inflammatory reaction secondary to local gentamicin administration.

This study had several limitations. First, the dilution effect was not estimated. The extent of ELF dilution depends on the specifics of the lavage procedure (volume instilled) relative to the donor airspace surface area. Several methods/calculations have been described to take into account the dilution effect. In bronchoalveolar lavage fluid, the ratio of plasma to bronchoalveolar lavage fluid urea or the albumin concentration has been described as an index of ELF dilution to determine the real concentrations of antibiotics or biomarkers [14,34,35,36]. In this study, the dilution effect of NAL was not taken into account, since the measured gentamicin concentrations were all far above the MIC. Moreover, we used an identical NALF procedure in each dog, and in view of the fact that all dogs were anatomically comparable, the comparison was considered valid. As a result, our values are probably much lower than the real concentrations in the ELF, but they could be adequately used to compare all three protocols. Secondly, this study failed to provide any data related to the pharmacoavailability of gentamicin over time in the nasal cavity and in the serum. This is, however, not a challenging issue since the efficacy of gentamicin against antimicrobials depends on the concentration and not time [9,10,11,12,13]. This study did not investigate the distribution of gentamicin in the nasal cavity. Although the drop protocol delivered a high gentamicin concentration, it is likely that it did not reach all the surfaces of the nasal cavity. Thirdly, the LOQ of 0.5 µg/mL was higher than the value obtained in many of the serum samples in the present study, which could reduce the adequacy of the statistical analyses. Finally, this study was performed in healthy dogs and not in dogs with sinonasal disease, meaning that the dynamics of the mucosal absorption of topically administered antibiotics might have been reduced by the lack of inflammation of the mucosa. In fact, in a disease state, the degree of translocation through the mucous membrane, mucous secretion and/or sneezing might positively or negatively affect gentamicin’s pharmacoavailability. This could also impact the distribution of gentamicin within the sinonasal cavity and in the peripheral circulation.

## 5. Conclusions

The three topical administration protocols (10-min protocol, 3-min protocol and drop protocol) achieved an efficient concentration against aminoglycoside-sensitive bacteria in NALF while remaining below the toxic values in serum. These three topical administration methods are interesting and safe alternatives to systemic antibiotics to control bacterial surinfection in sinonasal disorders. The comparison of the two inhalation protocols shows that the 3-min protocol allowed a sufficient C_MAX_ concentration in NALF together with a lower concentration in serum and a more interesting NALF/serum concentration ratio. However, the drop protocol, which was also the easiest one to administer, provided the highest concentration in NALF together with the lowest concentration in serum and thus can be recommended as the method of choice. Further studies are needed to assess the efficacy and safety of such therapeutic protocols in dogs with acute and chronic rhinosinusitis.

## Figures and Tables

**Figure 1 vetsci-10-00490-f001:**
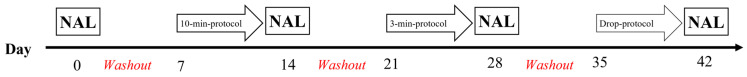
Study protocol presentation. NAL, nasal lavage; 10-min protocol, 10 min of gentamicin inhalation twice daily for 1 week; 3-min protocol, 3 min inhalation of gentamicin twice daily for 1 week; drop protocol, administration of 12.5 mg of gentamicin in each nostril twice a day for 1 week.

**Figure 2 vetsci-10-00490-f002:**
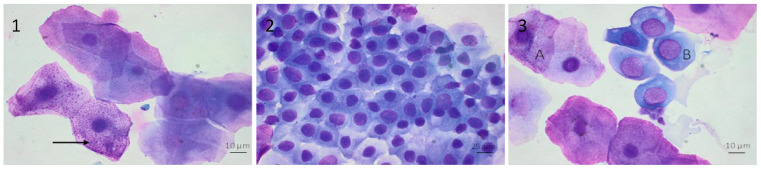
(**1**): Squamous cells in nasal lavage fluid. Characterized by large cells with polyhedral and angular appearance, the nucleus can still be present, pycnotic (<15% of the cell diameter), presence of keratohyalin granules in the cytoplasm in some cells (black arrow). (May–Grünwald–Giemsa stain × 100). (**2**): Epithelial cells in nasal lavage fluid. Characterized by moderate diameter, rounded shape with a large, rounded nucleus (nuclear diameter ± 30% of cells diameter) and little cytoplasm. (May–Grünwald–Giemsa stain × 40). (**3**): Keratinized squamous (A) and epithelial (B) cells in nasal lavage fluid. (May–Grünwald–Giemsa stain × 100).

**Figure 3 vetsci-10-00490-f003:**
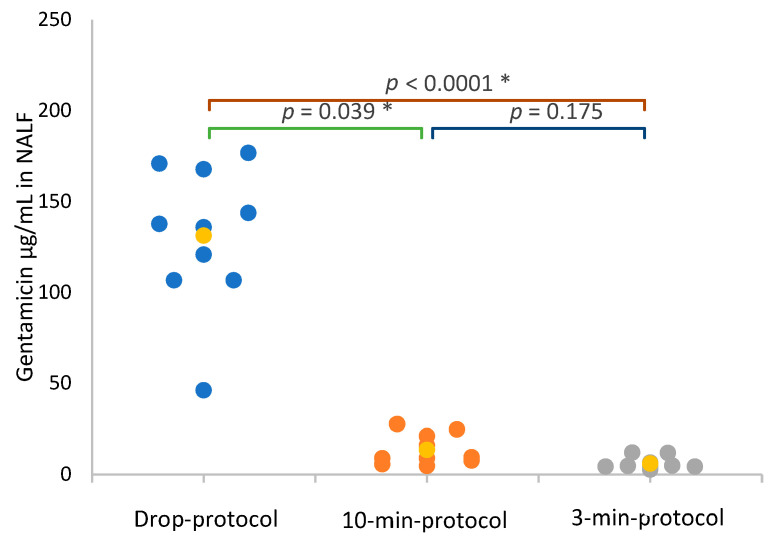
Scattergram of gentamicin concentration in the nasal lavage fluid, expressed in µg/mL, in the 3 groups: 10-min protocol, 3-min protocol and drop protocol. Yellow value corresponds to median value of each group. LOQ of gentamicin was 0.5 µg/mL, values below LOQ were assimilated as LOQ/2 (0.25 µg/mL). 10-min protocol: inhalation of gentamicin 5% for 10 min, twice daily, for 1 week. 3-min protocol: inhalation of gentamicin for 3 min, twice daily, for 1 week. Drop protocol: administration of 0.25 mL of gentamicin in each nostril, twice daily, for 1 week. LOQ: lower limit of quantification. *: *p*-value lower than 0.05 (significant). NALF: nasal lavage fluid.

**Figure 4 vetsci-10-00490-f004:**
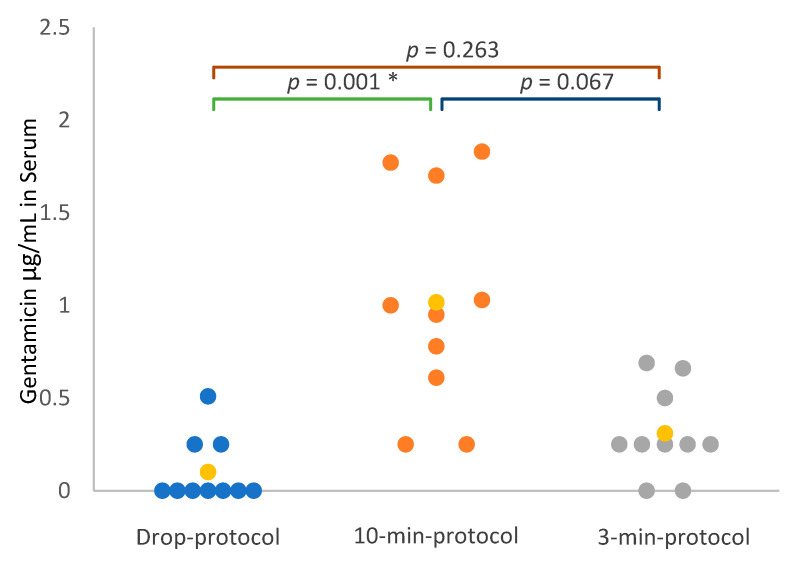
Scattergram of gentamicin concentration in the serum, expressed in µg/mL, in the 3 groups: 10-min protocol, 3-min protocol and drop protocol. Yellow value corresponds to median value of each group. LOQ of gentamicin was 0.5 µg/mL, values below LOQ were assimilated as LOQ/2 (0.25 µg/mL). 10-min protocol: inhalation of gentamicin 5% for 10 min, twice daily, for 1 week. 3-min protocol: inhalation of gentamicin for 3 min, twice daily, for 1 week. Drop protocol: administration of 0.25 mL of gentamicin in each nostril, twice daily, for 1 week. LOQ: lower limit of quantification. *: *p*-value lower than 0.05 (significant).

## Data Availability

No new data were created or analyzed in this study. Data sharing is not applicable to this article.

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
