# Peer review of "Intranasal and Serum Gentamicin Concentration: Comparison of Three Topical Administration Protocols in Dogs"

_vetsci, 2023, doi:10.3390/vetsci10080490_

Round 1

Reviewer 1 Report

This article was a prospective crossover study aimed at evaluating the concentration of nasal epithelial lining fluid and serum gentamicin in 3 topical administration protocols in 10 dogs. The study ultimately found that the drop protocol resulted in the highest concentration of gentamicin in the nasal cavity when compared with the nebulization protocol. The study processes are relatively standard, but the conclusions about the efficacy of these 3 topical administration protocols are very unreliable and arbitrary. The authors were trying to make this study more significant seemingly. There were multiple grammatical errors, first person pronouns, and personal opinions. The article needs to be revised extensively to improve the clarity and objectivity.

Title: No suggestion

Abstract and simple summary

1.     The two “during” in the sentence should be changed to “for a duration of”. (Line 14)

2.     The first appearance of “NALF” in the abstract needs to be clarified with its full name, such as "nasal lavage fluid (NALF)". (Line 17)

3.     In statistics, “P” is typically italicized, such as “P<0.05”. Additionally, it is common to use “P=0.039” instead of “P<0.039”. (Line 22)

4.     The conclusions about “10-min-protocol, 3-min-protocol and 25 Drop-protocol achieved sufficient concentration to have good efficacy against aminoglycoside sensible bacteria” are so arbitrary. There is nothing about the experiments of the efficacy of these 3 protocols in the whole paper. (Line 25-27)

5.     The words of “as the most suitable” is so exaggerated. (Line 34)

6.     The key words are too broad and should be summarized to make them more precise. For example, “upper respiratory tract disease” can be replaced by “nasal disease”, “topical therapy” and “protocol” might be combined as “topical administration protocol”. (Line 36)

Introduction

7.     The subsequent paragraphs mention the risks of toxicity and the greater precision of intranasal administration. It is recommended to clarify this point in this sentence. (Line 49-50)

8.     It may be considered to briefly describing the research progress of other drug nasal delivery routes, especially in dogs. (Line 68-70)

Materials and methods

9.     The four sample collection timepoints are confusing. Please clarify it by rephrasing it, such as “before one week washout (day 0), and after the final administration of each protocol (day 14, day 28, day 42).

10.   It is recommended to state the atomization procedure to determine whether the drug is effectively inhaled. (Line 99)

11.   “12,5 mg” or “12.5mg”? Please reconfirm and check the full text for other dose-related information. (Line 104)

12.   Recommended addition: Gentamicin was administered at the same concentration (50±0.01mg/mL) for 3 different protocols to reduce the effect of initial dose on drug concentration. (Line 102-105)

13.   Please check whether the statement is wrong. “20±1mL” or “20±1mL/kg”? (Line 116)

14.   Please elaborate on the purpose of conducting cytologic examination in the introduction part, such as cytology of the NALF samples can provide insights into the inflammatory process or altered pulmonary mechanics developed after the antimicrobial administration. (Line 123)

Result

15.   The time of the whole NAL procedure should also be stated as a range and the median, consistent with other data, and “±30 sec” here is not so clear. (Line 157)

16.   The sentence about “TCC of NALF ranged from 0 to 60 cells/μL at Day 1 (10 cells/μL [0-20], median [interquartile range 25-75])” is confusing. Please make it clear. (Line 166-167)

17.   Only the TCC data from day 1 was available. The TCC and differential cell count date was lack of the day 14, 28 and 42. The cytologic examination of different timepoints can be valuable in determining the efficacy and safety of nasal administration of gentamicin.

18.   Consider combining figures 2 through figure 4 in a single figure. (Line 176-194)

19.   It seems that the mark “A” and “B” are reversed in figure 4. (Line 191-192)

20.   The figure notes should be more concise. (Line 214-221)

Discussion

21.   The statement “Therefore, we are confident that concentrations obtained after the 3 protocols are effective against gentamicin sensitive bacteria growing in the nasal cavities” may be a little arbitrary. It is important to recognize the limitation in intranasal antimicrobial administration including its restricted reach, variable distribution, and the need for further research to determine its the therapeutic outcome. (Line 254)

22.   “0,42 μg/mL” or “0.42”? Please reconfirm and check the full text for other dose-related information. (Line 266)

23.   It’s not reasonable to assume that “the gentamicin levels measured in the serum of dogs in the present study are close to a peak serum concentration and do not exceed toxic values”. Please rephrase it. Or to gain a comprehensive assumption, it’s recommended to conduct multiple sample collections at different time intervals after the final administration. (Line 270)

24.   Please fix the grammar and improve the clarity of this sentence “However, the diagnostic and prognostic interest of NALF analysis in dogs with nasal disorders have to be determined, one of the major expected complications being the contamination of the NALF with bleeding.” (Line 308)

25.   The significance of NLF cell count and differentiated cell count was not discussed.

Conclusions

26.   The sentence about “... especially in case of multidrug resistant bacteria” cannot be a conclusion drawn from this study. (Line 341)

27.   The meaning of the parameter “NALF/serum concentration ratio” is not defined in the preceding part of the article. A clear definition or interpretation of its significance should be made. (Line 344)

28.   The different total dosage in three protocols might also be discussed as a drawback, because the gentamicin concentration between the groups were compared in this study. (Line 335)

Abbreviations

29.   The full name of TCC is incomplete. TCC: total cell count. (Line 361)

References

30.   The formats of references are not consistent.

Moderate editing of English language required

Author Response

REVIEWER COMMENTS:

Reviewer: 1

This article was a prospective crossover study aimed at evaluating the concentration of nasal epithelial lining fluid and serum gentamicin in 3 topical administration protocols in 10 dogs. The study ultimately found that the drop protocol resulted in the highest concentration of gentamicin in the nasal cavity when compared with the nebulization protocol. The study processes are relatively standard, but the conclusions about the efficacy of these 3 topical administration protocols are very unreliable and arbitrary. The authors were trying to make this study more significant seemingly. There were multiple grammatical errors, first person pronouns, and personal opinions. The article needs to be revised extensively to improve the clarity and objectivity.

Thank you very much for your careful review. The paper has been revised accordingly. We hope that these modifications will improve the paper clarity and objectivity.

Abstract and simple summary

  1. The two “during” in the sentence should be changed to “for a duration of”. (Line 14) Correction made in the text.
  2. The first appearance of “NALF” in the abstract needs to be clarified with its full name, such as "nasal lavage fluid (NALF)". (Line 17) Correction made in the text.
  3. In statistics, “P” is typically italicized, such as “P<0.05”. Additionally, it is common to use “P=0.039” instead of “P<0.039”. (Line 22) Correction made in several places in the text.
  4. The conclusions about “10-min-protocol, 3-min-protocol and 25 Drop-protocol achieved sufficient concentration to have good efficacy against aminoglycoside sensible bacteria” are so arbitrary. There is nothing about the experiments of the efficacy of these 3 protocols in the whole paper. (Line 25-27)

The sentence has been modified: “This study reveals that each of the 10-min-, 3-min- and Drop-protocols achieved concentration of gentamicin in NALF superior than the minimum inhibitory concentration for gentamicin-sensitive bacteria while remaining below the toxic values in blood.”

  1. The words of “as the most suitable” is so exaggerated. (Line 34) This section was removed.
  2. The key words are too broad and should be summarized to make them more precise. For example, “upper respiratory tract disease” can be replaced by “nasal disease”, “topical therapy” and “protocol” might be combined as “topical administration protocol”. (Line 36) Correction made in the text.

Simple summary was revised

Introduction

  1. The subsequent paragraphs mention the risks of toxicity and the greater precision of intranasal administration. It is recommended to clarify this point in this sentence. (Line 49-50)

We clarified the sentence: In general practice, dogs with nasal disease are often treated with systemic courses of oral non targeted antibiotics as first intention. However, repeated empirical administration of systemic oral antibiotics is related to development and spread of drug resistant microorganisms [8]. Increased prevalence of antibiotic resistance is a major concern in recent decades [8]. (…) Topical hydrophilic antibiotics such as aminoglycosides are of particular interest in order to reduce bacterial resistance [8].”

  1. It may be considered to briefly describing the research progress of other drug nasal delivery routes, especially in dogs. (Line 68-70)

we have added this paragraph : “In dogs the most commonly described intranasal delivery methods relate to nebulization of steroids (Bexfield JSAP 2006, Canone AM et al JSAP 2016)  or antibiotics (Morgane Canone 2020) for management of lower airway disease. Since nasal cavity is a suitable target for systemic delivery of drugs due to a large mucosal surface area combined with extensive vascularization providing an optimal absorption surface for drug delivery (Erdo et al 2018), intranasal delivery of drugs , such as  sedatives (lopez-ramis,)or vaccines (Ellis JA and al Comparative effects of intranasal and oral.. Vet J 2016) is also used to induce robust and prompt systemic effect. However local administration of antibiotics is not commonly used in dogs for treatment of upper respiratory tract diseases

Materials and methods

  1. The four sample collection timepoints are confusing. Please clarify it by rephrasing it, such as “before one week washout (day 0), and after the final administration of each protocol (day 14, day 28, day 42).

We propose: “In each dog, a NAL procedure was performed under anesthesia at four different timepoints. The first NAL procedure was performed before gentamicin administration at day 0. The three further NAL procedures were performed following on week gentamicin administration using three different topical gentamicin administration protocols at day 14, day 28 and day 42. Each protocol was separated by a one week wash out (Figure. 1). “

  1. It is recommended to state the atomization procedure to determine whether the drug is effectively inhaled. (Line 99)- we added a sentence.
  2. “12,5 mg” or “12.5mg”? Please reconfirm and check the full text for other dose-related information. (Line 104)- Correction made in the text.
  3. Recommended addition: Gentamicin was administered at the same concentration (50±0.01mg/mL) for 3 different protocols to reduce the effect of initial dose on drug concentration. (Line 102-105). We added a sentence, thank you.
  4. Please check whether the statement is wrong. “20±1mL” or “20±1mL/kg”? (Line 116). Correction made in the text.
  5. Please elaborate on the purpose of conducting cytologic examination in the introduction part, such as cytology of the NALF samples can provide insights into the inflammatory process or altered pulmonary mechanics developed after the antimicrobial administration. (Line 123)

We added this paragraph: Nasal lavage (NAL) collection has been used in humans to investigate and monitor rhinitis. This method is considered reliable and a less invasive/ irritative method compared with the current other diagnosis techniques as histopathology. Futhermore, this technique has been described as collection method in epidemiologic and experimental studies). In authors’ knowledge, reproducibility and description of NAL fluid (NALF) collection protocol have not been described in dogs.”

Result

  1. The time of the whole NAL procedure should also be stated as a range and the median, consistent with other data, and “±30 sec” here is not so clear. (Line 157)- correction made in the text.
  2. The sentence about “TCC of NALF ranged from 0 to 60 cells/μL at Day 1 (10 cells/μL [0-20], median [interquartile range 25-75])” is confusing. Please make it clear. (Line 166-167) – we modified the sentence: “At day 0, NALF total cell count ranged from 0 to 60 cells/µL (10 cells/µL [0 - 20], median [interquartile range 25 - 75]). Three types of cells were observed: squamous epithe-lial cells (50% [38 - 52]), epithelial cells (48% [48 - 55]) and neutrophils (2% [0 - 4]).”
  3. Only the TCC data from day 1 was available. The TCC and differential cell count date was lack of the day 14, 28 and 42. The cytologic examination of different timepoints can be valuable in determining the efficacy and safety of nasal administration of gentamicin.

We essentially wished to describe it in healthy animals, allowing for comparison with sick animals in future studies. However, we did measure the TCC and DCC at days 14, 28 and 42 , which did not  differ significantly from baseline.  “There were no significant differences in NALF total cell count and differential cell counts between all timepoints. After 10-min-protocol, NALF total cell count was 8 cells/µL [0 - 18]. Differential cell count included epithelial cells: 58 % [48 - 63], squamous epithelial cells: 40 % [32 - 52]) and neutrophils 2% [0 - 4]. After 3-min protocol, NALF total cell count was 10 cells/µL [0 - 18]. Differential cell count included epithelial cells 62 % [48 - 65], squamous epithelial cells 48 % [33 - 54] and there were no neutrophils. After drop protocol, NALF total cell count was 15 cells/µL [0 - 19]. Differential cell count included epithelial cells in 50 % [40 - 65], squamous epithelial cells 49 % [35 - 55] and neutrophils 1% [0 - 1]. “

  1. Consider combining figures 2 through figure 4 in a single figure. (Line 176-194) – correction made in the text.
  2. It seems that the mark “A” and “B” are reversed in figure 4. (Line 191-192) – correction made in the text.
  3. The figure notes should be more concise. (Line 214-221) - correction made in the text.

Discussion

  1. The statement “Therefore, we are confident that concentrations obtained after the 3 protocols are effective against gentamicin sensitive bacteria growing in the nasal cavities” may be a little arbitrary. It is important to recognize the limitation in intranasal antimicrobial administration including its restricted reach, variable distribution, and the need for further research to determine its the therapeutic outcome. (Line 254). We modified the sentence. “Therefore, we expect that concentrations obtained after the 3 protocols might be effective against gentamicin sensitive bacteria growing in the nasal cavities.”
  2. “0,42 μg/mL” or “0.42”? Please reconfirm and check the full text for other dose-related information. (Line 266) - correction made in the text.
  3. . It’s not reasonable to assume that “the gentamicin levels measured in the serum of dogs in the present study are close to a peak serum concentration and do not exceed toxic values”. Please rephrase it. Or to gain a comprehensive assumption, it’s recommended to conduct multiple sample collections at different time intervals after the final administration. (Line 270). We rephrased it. “In man, the peak serum concentration after nasal irrigation is described to occur after 30 to 60 min [20]. Therefore, and although we measured the concentration of gentamicin in serum only once within that period time (35 min after drug administration), we presume that the gentamicin levels measured in the serum of dogs in the present study are close to a peak serum concentration and do not exceed toxic values.
  4. Please fix the grammar and improve the clarity of this sentence “However, the diagnostic and prognostic interest of NALF analysis in dogs with nasal disorders have to be determined, one of the major expected complications being the contamination of the NALF with bleeding.” (Line 308)

We modified the sentence: “However, the diagnostic and prognostic interests of NALF analysis in dogs with nasal disorders have to be determined. Increased number of inflammatory cells, or increased proportion of one type of inflammatory cells, detection of fungal and/or bacterial agents or of tumor cells could help in the diagnosis of nasal pathologies. One of the major expected complications would be the contamination of the NALF with blood, making proper interpretation difficult if not impossible In the present study TCC and differential cell count did not differ between the different timepoints, suggesting absence of nasal inflammatory reaction secondary to local gentamicin administration.”

  1. The significance of total cell count and differentiated cell count was not discussed.

There is a paragraph in the section discussion related to the NAL procedure and TCC and differential cell count: “It showed that in healthy dogs, NALF has a poor total cellularity and is mainly composed of squamous and epithelial cells populations. These results are consistent with the composition of the nasal epithelium, which is a multi-layered nasal and keratinized pigmented epithelium, and are also consistent with findings in human medicine [28]. » We added a section on the TCC obtained at days 14-28-42.

Conclusions

  1. The sentence abut “... especially in case of multidrug resistant bacteria” cannot be a conclusion drawn from this study. (Line 341). Thank you, we removed the sentence.
  2. The meaning of the parameter “NALF/serum concentration ratio” is not defined in the preceding part of the article. A clear definition or interpretation of its significance should be made. (Line 344). We added/ modified several sentences: “The ratio between the NALF and serum gentamicin concentrations (NALF to serum concentration ratio) was then calculated for each paired measurement as a reflexion of systemic drug delivery” (…) Finally, the calculated gentamicin concentration NALF/serum ratio was below 20 in the inhalation protocols (at 13 in the 10-min-protocol and 19 in 3-min-protocol) while it was above 1000 in the Drop protocol. (…) The higher NALF gentamicin /serum ratio found in drop protocol compared with both inhalation protocols show that this protocol may be the most appropriate.
  3. The different total dosage in three protocols might also be discussed as a drawback, because the gentamicin concentration between the groups were compared in this study. (Line 335)

We don't think this is a limitation, as one of the goals of the study was to compare 3 therapeutic set ups, resulting by essence in different total dosages.

Abbreviations

  1. The full name of TCC is incomplete. TCC: total cell count. (Line 361). Thank you, correction made in the text.

References

  1. The formats of references are not consistent. Correction made in text.

Reviewer 2 Report

This study was investigated to compare the concentrations of gentamicin in NALF and serum after three topical protocols and suggested that the drop-protocol was faster and better tolerated compared to the two inhalation protocols.

Experimental limitations have been suggested for the application of the research results to actual veterinary clinical practice, but additional supplementation is required.

Additional discussion is needed on the variables due to changes in blood concentration (degree of translocation through the mucous membrane, change in mucous secretion, etc.) for topical application in an infection or disease state.

Author Response

REVIEWER 2

Comments and Suggestions for Authors

Thank you for your review.

This study was investigated to compare the concentrations of gentamicin in NALF and serum after three topical protocols and suggested that the drop-protocol was faster and better tolerated compared to the two inhalation protocols. Actually all protocols were well tolerated (the drop protocol was not better tolerated, it was just straightforward did not require any equipment). We have changed the last sentence of the discussion accordingly

Experimental limitations have been suggested for the application of the research results to actual veterinary clinical practice, but additional supplementation is required. Do you mean that we need to expand this section according to the remark below? or other limitations?

Additional discussion is needed on the variables due to changes in blood concentration (degree of translocation through the mucous membrane, change in mucous secretion, etc.) for topical application in an infection or disease state. We have now expanded  the discussion at the end of the section “limitations”. A new sentence has been added: “ Finally, this study was performed in healthy dogs and not in dogs with sinonasal disease, meaning that the dynamics of mucosal absorption of topically administered antibiotics might have been reduced by the lack of inflammation of mucosa. In fact, in dis-ease state, degree of translocation through the mucous membrane, mucous secretion and/or sneezing might positively or negatively affect gentamicin pharmacoavailability. This can also impact the distribution of gentamicin within the sinonasal cavity as well as in the peripheral circulation.”